# Entropy and the Brain: An Overview

**DOI:** 10.3390/e22090917

**Published:** 2020-08-21

**Authors:** Soheil Keshmiri

**Affiliations:** The Thomas N. Sato BioMEC-X Laboratories, Advanced Telecommunications Research Institute International (ATR), Kyoto 619-0237, Japan; soheil@atr.jp

**Keywords:** entropy, differential entropy, permutation entropy, multiscale entropy, transfer entropy, brain function, brain connectivity, neuroscience, consciousness, ageing, hypnosis, anaesthesia, anaesthetic drug

## Abstract

Entropy is a powerful tool for quantification of the brain function and its information processing capacity. This is evident in its broad domain of applications that range from functional interactivity between the brain regions to quantification of the state of consciousness. A number of previous reviews summarized the use of entropic measures in neuroscience. However, these studies either focused on the overall use of nonlinear analytical methodologies for quantification of the brain activity or their contents pertained to a particular area of neuroscientific research. The present study aims at complementing these previous reviews in two ways. First, by covering the literature that specifically makes use of entropy for studying the brain function. Second, by highlighting the three fields of research in which the use of entropy has yielded highly promising results: the (altered) state of consciousness, the ageing brain, and the quantification of the brain networks’ information processing. In so doing, the present overview identifies that the use of entropic measures for the study of consciousness and its (altered) states led the field to substantially advance the previous findings. Moreover, it realizes that the use of these measures for the study of the ageing brain resulted in significant insights on various ways that the process of ageing may affect the dynamics and information processing capacity of the brain. It further reveals that their utilization for analysis of the brain regional interactivity formed a bridge between the previous two research areas, thereby providing further evidence in support of their results. It concludes by highlighting some potential considerations that may help future research to refine the use of entropic measures for the study of brain complexity and its function. The present study helps realize that (despite their seemingly differing lines of inquiry) the study of consciousness, the ageing brain, and the brain networks’ information processing are highly interrelated. Specifically, it identifies that the complexity, as quantified by entropy, is a fundamental property of conscious experience, which also plays a vital role in the brain’s capacity for adaptation and therefore whose loss by ageing constitutes a basis for diseases and disorders. Interestingly, these two perspectives neatly come together through the association of entropy and the brain capacity for information processing.

## 1. Introduction

Over the course of last two decades, neuroscientific research has accumulated comprehensive empirical [1,2,3] and theoretical [4,5,6] evidence for the crucial role of the brain signal variability in its function. This variability has been shown to emerge from the interplay between individual neurons and their neuronal circuits [7,8] and to extend over wide spatiotemporal scales in the brain [9,10]. Substantial findings relate its occurrence to the brain self-organized criticality [11,12,13,14,15] in which the brain maximizes its information capacity [16,17].

These findings, in turn, help explain the pivotal role of entropy (Appendix A) in quantification of the brain’s information processing [18,19,20,21], given the direct correspondence between the variance and the amount of information [22]. Today’s applications of entropy for quantification of the brain function range from information capacity of working memory (WM) [23] and the neural coding [24,25] to the interplay between neural adaptation and behaviour [26], functional interactivity between the brain regions [27], and the state of consciousness [28,29].

Entropy has also manifested as a robust feature for classification of the individuals’ mental states in such diverse domains as detection of the onset of epileptic seizure [30] and the change in state of vigilance [31]. For instance, Zheng and Lu [32] showed that the use of entropy for emotion classification based on electroencephalography (EEG) time series outperformed such feature spaces as differential asymmetry (DASM), rational asymmetry (RASM), and power spectral density (PSD). Similarly, Keshmiri et al. [33] verified the discriminative power of entropy in comparison with the dominant feature spaces in near-infrared spectroscopy (fNIRS) analysis of the n-back WM task [34]. Most recently, Liu and colleagues [35] used a considerably large resting-state functional magnetic resonance imaging dataset from the Human Connectome Project (998 individuals). They demonstrated that the information from cortical entropy profiles could effectively predict diverse facets of human subjects’ cognitive ability. In fact, there is ample evidence for underlying relations between the entropic measures on the one hand and the (brain) signal variability on the other hand [36,37,38,39,40,41,42,43,44,45]. These studies have advanced our understanding of the practical considerations while using these measures [46,47,48,49]. This, in turn, has paved the path for development of information-theoretic software and libraries [50,51,52,53,54,55,56,57,58] that facilitated the deployment of these measures for the study of the brain.

In the past, several reviews have summarized the use of entropic measures in neuroscience. For instance, Takahashi [59] reviewed the literature on the utility of measuring neurophysiological complexity associated with mental disorders in light of shared neurobiological conditions. However, this review was not primarily based on the use of entropic measures and instead covered different nonlinear analytical methodologies. Borst and Theunissen [25] highlighted the pros and cons of the use of information theory, in general, for capturing the information-content of neural responses to stimuli. Similarly, Quiroga and Panzeri [20] showed how decoding methodologies may yield further insights on neuronal population responses through the adaptation of information theory. However, their review primarily centred around the application of information theory in decoding strategies pertinent to single-trial population analyses. On the other hand, Carhart-Harris [18] and Carhart-Harris et al. [60] considered entropy as a means of studying the state of consciousness in relation with the effect of psychedelic drugs. The authors provided a comprehensive coverage of the related literature which formed a solid foundation for their hypothesis: the “entropic brain” hypothesis that proposes the entropy as a marker of the conscious states in spontaneous brain activity.

### The Present Overview

The present overview aims at complementing the aforementioned reviews by focusing on three domains of research: the (altered) state of consciousness, the ageing brain, and the quantification of the brain networks’ information processing. The reasons why these three areas of research are covered by the present overview are twofold. First, the study of consciousness, the ageing brain, and the brain networks’ information processing are among the most active fields of research, to the best of our knowledge, where the use of entropy has resulted in highly promising findings. Second (and perhaps more importantly), the use of entropy in these areas helped realize that their seemingly different lines of research are indeed highly intertwined and related to each other. On the one hand, a number of theses consider complexity as a fundamental property of conscious experience [61,62] where the evolution of brain’s complexity manifests as an entropy-enhancing process [63]. On the other hand, there is growing evidence that identifies the vital role of physiologic complexity in organisms’ capacity for adaptation [26] and pp. 3, 93–94 in [64], thereby relating the age-related loss of complexity with diseases and disorders [59,60,65,66,67]. Interestingly, these two propositions substantially overlap through their association with the brain capacity for information processing [13,68,69,70].

In the case of (altered) states of consciousness, the present overview highlights the utility of entropy for differentiating the brain activity in three states of wakefulness, anaesthesia, and unresponsive wakefulness syndrome. It also summarizes the findings that show the effectiveness of entropy for quantification of the brain function during hypnosis. In the case of the ageing brain, it brings together the studies that identify the correspondence between the brain complexity and ageing and how such a relation translates into the change in the brain function with ageing. For the case of quantification of the brain networks’ information processing, it summarizes the findings that shed further lights on the functional connectivity studies in terms of brain networks’ complexity and information processing.

There were two ways to organize the sections of this overview: (1) to focus on types of entropic measures (2) to categorize their contents as per domains of application. The first choice would allow for highlighting the different formulations of entropic measures and their subsequent modifications. However, it could also make it more inconvenient for the readers to pinpoint the fields of research in which these measures yielded substantial results. On the other hand, the second option would allow for more tangibly bringing together the incremental findings in each area of research, while putting less emphasis on types of entropic measures that were utilized. To more efficiently appreciate the recent progress based on the application of entropy in neuroscientific studies, the present overview adapted the second approach. For interested readers, it also provided a brief summary of three most commonly used entropic measures in Appendix B. These measures are (differential) entropy (DE) [48,71], multiscale entropy (MSE) [72], and permutation entropy (PE) [73].

Last, the use of entropy for quantification of the interplay between psychedelic drugs and consciousness is an exciting venue in which the neuroscientific research has yielded interesting and promising findings. However, the present study did not cover this topic. The interested readers may find a comprehensive coverage of this topic in Carhart-Harris [18] and Carhart-Harris et al. [60].

## 2. (Altered) State of Consciousness

Classical approaches to the study of consciousness primarily focus on contribution of specific brain areas or groups of neurons [62]. However, such approaches fall short in realizing that the modulation of information does not necessarily involve a change in brain’s regional power and/or activity [74]. Such a shortcoming can be addressed by analyzing the brain’s activity using entropic measures that are, by principle, designed for quantification of the amount of information. This, in turn, allows for identification of the brain signal’s complexity (Appendix A) as a fundamental property of conscious experience.

### 2.1. Various Stages of (Un)consciousness

King et al. [75] demonstrated that the state of consciousness was associated with markedly increased information sharing among distributed brain regions. They further identified that such an increase in brain distributed information was particularly observable over medium- and long-range brain connectivity. Interestingly, the degree of change in such a distributed information sharing was a reliable marker to robustly distinguish between unresponsive wakefulness syndrome, minimally conscious state, and conscious state. These results complemented the findings on the relation between information sharing and loss of consciousness that were reported using spectral-based functional connectivity measures [76,77]. Furthermore, and unlike these synchrony-based measures, their approach allowed for minimization of the effect of common-source artefacts, thereby robustly differentiating between distinct states of consciousness. Their findings also provided substantial evidence for a number of theoretical models of consciousness that envisioned the brain’s large-scale information sharing as a consistent signature of conscious processing [62,78,79,80,81]. The authors posited their approach may enhance the current behavioural and neuroimaging methodologies to more effectively diagnose consciousness-related disorders [82].

Zhang et al. [29] used the entropy of Markov trajectories [83] to demonstrate that human consciousness relies on the temporal circuit in which the dynamic is characterized by the balanced and reciprocal accessibility of the default mode network (DMN) [84] and the dorsal attention network (DAT) [85]. Their findings provided further support for involvement of these two brain networks as two distinct cortical systems that support consciousness [85,86,87,88]. These findings further explained the diminished anti-correlation between these two networks during unconsciousness [89,90,91] in terms of spatiotemporal brain dynamics. These findings further advanced the study of the conscious brain by revealing that the disruption of this spatiotemporal brain dynamic might be the common signature of various forms of unresponsiveness.

Demertzi and colleagues [28] further advanced these previous findings by identifying the presence of conscious brain dynamics in sustained patterns of long-range coordination. These patterns showed a low similarity to the anatomical connectivity. Their analyses also associated the reduced/absence of conscious processing to brain dynamics that exhibited a low interregional coordination. Such alternating states of high and low brain dynamics has been known to constitute a basis for complex cognitive functions in which integrated states support faster and more accurate cognitive performance [92]. They were also in accord with the theories that relate the state of consciousness to a brain-scale activity in which self-sustained coordination [93,94,95] allows for manifestation of perception, emotion, and cognition [96,97,98]. Their findings further extended the previous research on nonhuman primates that showed the anaesthetized brain activity mostly resembled the brain anatomical connectivity patterns [99] to the case of humans. They also provided empirical evidence for the theoretical stance that suggested such alternating patterns of brain dynamics may constitute a fundamental property of the brain information processing [87].

Miskovic et al. [100] studied the change in human brain signal variability during the sleep cycle and found that the brain signal entropy throughout the sleep cycle was strongly time-scale dependent. Their results indicated that the slow wave sleep (SWS) was associated with a reduced complexity at short time scales which was accompanied with its increase in long time scales. They further noted that the temporal signal complexity at short time scales and the slope of the EEG power spectral captured a neuronal noise that potentially reflected the cortical balance between excitation and inhibition [64]. These results established an explicit link between the brain’s entropy and the $\frac{1}{f}$ (Appendix C, Definition A2) component [36] of power spectra [101,102]. This, in turn, extended the previous research, which showed that the slope of power spectrum density (PSD) at large-scale field potentials acts as a measure of the neuronal population spiking [103,104]. They further verified the association between the smaller time scales’ entropy and the brain’s local information processing [105,106,107,108,109,110]. Additionally, they provided further support for the observation that the increase in sleep depth was accompanied by greater inhibitory activity [111] and the lowered global level of consciousness [112,113]. Another interesting observation in this study was the increased level of entropy in stage 3 non rapid-eye-movement (NREM-3) sleep compared to its increase in NREM-2 and REM at large time scales [114]. This suggested the switching of the cortical activity into a global bistable pattern of depolarized and hyperpolarized states that are the characteristics of SWS [108].

### 2.2. Anaesthesia

Olofsen et al. [115] observed that the change in entropy tracked the qualitative effect of anaesthetic drug from awake state to sedated/lightly anaesthetized and eventually the deep anaesthetized state. They further observed that, compared to other anaesthetic indices, their index did not require long segments of EEG data. Additionally, it required a minimal pre-processing of data and was highly resistant to blink artefacts while being computationally efficient. The authors also noted that the open-access nature of their index presented a reliable alternative to proprietary anaesthetic indices, the results of which could not be readily interpreted (due to unavailability of their algorithmic logic). These findings were further replicated by other studies [116,117,118].

### 2.3. Hypnosis

Hypnosis has received a growing interest from cognitive neuroscience research, in part, due to its utility for studying the consciousness [119,120]. To realize the effect of hypnosis on the brain, a number of biomarkers have been introduced. They include structural synchrony measure [121], Phase Lag Index [122], topographic variability in the beta and gamma bands [123], coherence (COH) [124] and the imaginary component of coherence (iCOH) [125] (for a brief description of brain wave’s frequency bands, see Appendix C, Definition A3). Although these measures provided encouraging results, their applicability appeared to be limited. Whereas Deivanayagi et al. [126] found that COH associated the state of hypnosis with lowered theta and alpha frequency bands, Sabourin et al. [127] found that COH indicated an increase in theta power during hypnosis in both low as well as high hypnotizable individuals. In the same vein, the use of COH and iCOH [128] were both required to analyze different frequency bands (i.e., one per theta and beta1 bands, respectively). Furthermore, they did not identify any significant differences in power [128].

Interestingly, entropy provided a potential unifying measure to reproduce these previous observations that were based on multiple biomarkers [129]. Precisely, it identified a large bi-hemispheric effect of hypnotic suggestibility (with comparable strength in their effect sizes) on information-content of theta, alpha, and beta frequency bands. This observation was in accord with Han et al. [130] who showed that the signals that were carried by individual cortical neurons concurrently contributed to multiple functional pathways in both hemispheres, thereby providing promising evidence in support of the global brain neural excitation in response to stimuli. Entropy also identified a significantly higher functional connectivity (FC) among higher hypnotic suggestible participants’ theta and alpha bands that was more pronounced in the parietal (in the case of theta) and centroparietal (for alpha) regions and that was accompanied by a non-significant smaller FC in the beta band in the central region. In this respect, Jamieson and Burgess [128] also reported similar changes in FC from pre-hypnosis to hypnosis state using iCOH (increase in theta) [125] and COH (decrease in beta1) [124]. However, their analyses, which were primarily based on the state of hypnosis (i.e., without observing responses of the participants to hypnotic suggestions), did not identify any significant differences [128] between pre-hypnosis and hypnosis state on these bands. It is also worthy of note that the entropy’s identification of the role of alpha and theta bands during the brain processing of the hypnotic suggestions were in line with the previous findings on the pivotal role of these bands in information processing and transfer of information between functionally connected brain regions [121,122,131,132,133,134].

## 3. The Ageing Brain

Section 2 identified the substantial contribution of the brain’s entropy in formation of a conscious experience. In the same vein, a growing number of recent studies also hint at the interplay between physiological complexity and the brain’s capacity for adaptation. This section highlights some of these findings that show the significant contribution of the brain’s signal complexity to its development and also recognize the loss of such a complexity through the process of ageing as a major contributing factor to various age-related cognitive declines and deficiencies.

### 3.1. The Interplay between Brain Signal Variability and Its Development

McIntosh et al. [135] showed that the brain signal variability increases by age. They also showed that such a variability correlated with the reduced behavioural variability and therefore resulted in more accurate performance in cognitive tasks. Their findings also identified the observed variability in the brain activity to be a critical feature of its function [3,7].

These findings advanced the previous research on developing brain by validating that the neural system maturation promotes an increased physiological variability, thereby allowing for better environmental adaptability [136,137]. More specifically, they pinpointed the vital role of the signal variability in enabling the brain to parse weak and ambiguous incoming signals [138,139,140]. This, in turn, verified the central role of the brain signal variability in facilitating such functions as inter-neuronal signal exchange [141,142], transitional states in metastable systems [143], and the formation of functional networks [144,145].

### 3.2. The Power-Law Scaling of the Brain Signal Variability in Adulthood

Takahashi and colleagues [146] studied the effect of photic stimulation on the brain activity of healthy younger and older adults. They identified a significant increase in the brain signal complexity that was only present among the younger individuals. This indicated that unlike older adults, the brain of younger individuals exhibited a power-law scaling property that corresponded to the long-range temporal correlation between their brain regions.

These findings extended the previous research on the effect of ageing on brain function in two ways. Second, the absence of power-law scaling in the older adults’ brain dynamics helped establish a relation between reduced/diminished brain ability to respond to an external stimuli with ageing [147,148]. Second, it identified that such a scaling that corresponds to the intrinsic complexity in physiological systems [65] to be also vital for the healthy brain functioning. Third, they further advanced the proposal [149] that the loss of complexity results in functional decline of the organism by diminishing the range of available, adaptive responses to events of everyday life [66].

### 3.3. Ageing and the Default Mode of the Brain

Yang and colleagues [150] demonstrated that, compared to younger adults, the older individuals’ default mode network (DMN) [84] exhibited a reduced complexity. They further verified that such a decrease in complexity was more significant in DMN’s posterior cingulate gyrus and hippocampal components.

These results contributed to the prior studies of the brain’s normal ageing in three distinct ways. First, they provided empirical support for the hypothesis that considers the ageing to underlie the reduced network’s complexity and information integration of the brain [151,152,153,154]. Second, they verified that the reduced brain complexity is at the core of the older adults’ significantly lower magnitude of DMN co-activation in the posterior cingulate cortex [155,156]. Third, they verified that the reduced network efficiency in older individuals’ frontotemporal and limbic brain regions [157] could be consequentially explained in terms of observed decrease in brain’s signal variability.

### 3.4. Ageing and the Brain’s Distributed Versus Local Information Processing

McIntosh and colleagues [109] studied the interplay between ageing brain’s local and distributed information processing using two independent datasets of EEG and magnetoencephalography (MEG) recordings. These datasets included both younger and older human subjects.

Their results indicated that whereas most brain regions exhibited an increase in local information processing, the change in distributed brain’s complexity was age-related and was, subsequently, reduced across hemispheres by ageing. More importantly, they observed that unlike early life maturation in which the changes in the brain complexity were more widespread, such changes during adulthood exhibited a strong spatiotemporal dependency.

An important implication of these findings was the attribution of the observed older adults’ higher brain modularity [158] as a direct consequence of its increased local information processing that could potentially be due to its reduced capacity for distributed information processing. Furthermore, the observed age-related temporal changes in the brain complexity in this study extended the prior results on the changes in spectral power during normal ageing [159,160] that identified a general decrease/increase in power in lower/higher brain frequency bands. In this regard, as a result, they extended the findings on increased brain regional long-range interactions and its reduced local complexity in early life [107] to the case of the ageing brain. Additionally, the predominantly cross-hemispheric nature of observed decreases in the complexity in their study complemented the coherence-based studies that noted the reduction in inter-hemispheric functional connections with age [161,162]. Taken together, the study by McIntosh et al. [109] supported the dedifferentiation hypothesis [67,163,164,165,166,167] which states that the age-related decline in the brain and behaviour and the loss in brain complexity are synonymous and integrated.

## 4. Quantification of the Brain Networks’ Information Processing

Section 2 and Section 3 highlighted the correspondence of brain’s entropy to manifestation of various states of conscious experience on the one hand and the decline of the brain’s function in terms of loss of such an inherent complexity through the process of ageing on the other hand. As the current section reveals, such a correspondence is not a coincident but a substantial relation that emerges from the association between the brain’s inherent complexity and its capacity for information processing.

### 4.1. Emergence of the Distinct Functional Networks

Through introduction of a novel entropic functional connectivity index, Tononi and colleagues [27] showed that certain subsets of brain regions may interact more strongly among themselves than with the rest of the brain. They further argued that such variations among differential brain’s sub-regions might be crucial for the emergence of the functional boundaries from widespread brain global connectivity [168].

These results further advanced the cognitive neuroscience studies by demonstrating that while many brain regions are active in the control of cognition and behaviour, only a subset of active neuronal groups is directly correlated with conscious experience [169,170].

### 4.2. Variability in the Brain’s Distributed Functional Networks

McDonough and Nashiro [110] found that the patterns of brain signal variability were distinct from noise. In addition, these patterns were differentially expressed among such distributed networks as DMN, cingulo-opercular network, and the left and right fronto-parietal (attention) networks [171] (Chapter 7, pp. 274–323). In particular, the complexity of these networks was negatively/positively associated with functional connectivity at small/large time scales (see Appendix B for details on time-scaled quantification of the brain signal variability through signal’s coarse-graining process). This suggested that the DMN might be accommodating a higher degree of information processing across distributed connections (e.g., medial frontal and medial parietal regions). It is worth noting that the authors quantified the functional connectivity using dual regression analyses [172,173] in which the regression weights were interpreted as a strength of the functional connectivity [174].

The contributions of these findings were threefold. First (and perhaps most importantly), they indicated that the brain signal variability was distinct from the noise [1,3,7,8]. Second, they showed that, relative to the other networks, DMN exhibited the smallest and largest degrees of network complexity at short and mid to large time scales relative to other networks. Third, they identified that the presence of such temporal scalings might be critical for understanding the dynamics of the inter-neuronal transfer of information [175]. This was in accord with the thesis that considers DMN’s regions to serve as critical gateways for information processing and integration within the local and distributed brain’s networks [176,177,178]. For instance, DMN has been related to episodic memory, imagining the future, self-reflection, mentalizing, divergent thinking, working memory, reading comprehension, and constructing moral judgments [171] (Chapter 13, pp. 515–565) and [179,180,181,182,183].

Additionally, the observed neural complexity in this study may also be interpreted as evidence for the presence of differential range or capacity of the brain for exploring the alternative brain states [3]. In fact, Wang and colleagues [184] found further evidence for these results and proposed that the complexity of the regional neural activity may serve as an index of the brain’s capacity for information processing. Specifically, they envisioned the presence of this increase in complexity to indicate the brain’s capacity for transitioning between its different states and networks to promote a greater propensity for information processing.

### 4.3. Variability and the Distributed Functional Synchrony

Liu et al. [185] observed (in mice under anaesthesia and wakefulness) a strong negative spatiotemporal linear correlation between functional connectivity and entropy. They also observed that entropy correlated positively with the complexity of the cortical activity in small time scales. This correlation was negative in large time scales. They further explored the validity of their results using simulated human’s brain activity (i.e., simulated blood oxygen level dependent, BOLD). Using this simulated data, they observed that the functional connectivity and complexity provided partial assessments (although with a reduced resolution) of the structural and dynamics variation of the cortical entropy.

This study contributed to the study of the brain function in terms of change in its signal variability in two substantial ways. First, it identified a scale-dependent correlation between complexity and entropy. Precisely, this finding verified that whereas the brain signal variability at small time scales was associated with its local information processing, it reflected the brain networks’ long range communication at larger scales [107,109]. Second, it demonstrated that anaesthesia affected the brain function by (1) reducing its entropy (2) strengthening its functional connectivity (3) decreasing/increasing its signal variability at small/large times scales. In this respect, these observations extended the previous sleep research that reported an increase in functional connectivity (within the dorsal attention network) during light sleep [186] and a global decrease in complexity at small/large time scales during deep sleep [100,114].

## 5. Concluding Remarks

The present overview sought to bring together three areas of neuroscientific research in which the use of entropy for quantification of the brain activity has yielded promising results. These areas were the (altered) state of consciousness, the ageing brain, and the quantification of the brain networks’ information processing. While doing so, it highlighted three main observations. First, it identified that the use of entropic measures for the study of consciousness and its (altered) states led the field to substantially advance the previous findings. For instance, they helped verify the theoretical models that envisioned the brain’s large-scale information sharing underlies its conscious processing [75], that DMN and DAT networks were crucially involved in such processes [29], and that the conscious brain’s dynamics sustained patterns of long-range coordination which was substantially distinct from its anatomical connectivity [28]. These studies further validated the utility of entropic measures as potential biomarkers for the study of differential states of consciousness [115,116,117,118,129]. Second, it realized that the use of these measures for the study of the ageing brain provided significant insights on various ways that the process of ageing may affect its dynamics and information processing capacity. For example, they identified that the brain signal variability was a critical feature of its function [135] in which loss by ageing may help explain its functional decline [146] and reduced information integration [109,150]. Third, it revealed that their utilization for analysis of the brain regional interactivity formed a bridge between the previous two research areas. This included the evidence for the emergence of functional boundaries from stronger interaction among subsets of brain’s global connectivity [27], the involvement of DMN in its higher information processing [110], and its reduced functional connectivity by ageing [185,187,188].

Collectively, these studies hinted at the brain’s complexity as its fundamental property that underlies the manifestation of such phenomena as consciousness [61,62] and adaptability [26] and pp. 3, 93–94 in [64]. This, in turn, resulted in attribution of various brain-related deficiencies and disorders [59,60,65,66,67] to the decline of this inherent complexity. These findings appeared to converge on the utility of brain’s complexity in its function and information processing [13,68,69,70].

The use of entropy for analysis of the brain regional interactivity also shed further light on the correspondence between the brain regional synchrony on the one hand and its dynamical complexity on the other hand. In this respect, Reid et al. [189] noted that associations among the brain regions in terms of correlation and synchronized activity can arise in a variety of ways that may not relate to the extent of the influence among these co-occurring processes. As a result, such measures may fall short of capturing a more comprehensive mapping between the observed associations and their underlying neural substrates [189,190,191]. These shortcomings can be addressed by the future research through utilization of such tools as Granger causality (GA) [192,193,194] and transfer entropy (TE) [19,195,196]. These tools can enable the research to devise analytical approaches that study the brain networks’ complexity in light of directed flow of information among their components. An advantage of TE compared to GA is that whereas the latter is based on linear vector autoregressive (VAR) [197,198] (and hence linear in nature), the former is a nonlinear directional measure of flow of information [41,199]. Therefore, its adaptation for the study of brain networks’ information processing and dynamics may yield more informative results.

In the same vein, the future research can further advance the study of (altered) states of consciousness and its relation with the brain complexity through utilization of such entropic frameworks as integrated information theory (IIT) [62,81,93,200]. This is a particularly interesting venue, considering the recent growth of our understanding about this measure’s interpretability [201]. This observation becomes more intriguing by the availability of toolboxes that substantially facilitate its computation [56,202] even for relatively large scale brain networks [203].

Another area of research for future exploration is the relation between entropy and neuropsychological disorders. In this regards, the review by Takahashi [59] found inconsistent results among studies that utilized nonlinear analytical methodologies for this purpose. As noted by the author, such studies can benefit from methodologies that take into account the brain signal complexity and its dynamics across multiple time scales [42,72]. Such studies can also benefit from the recent findings [35] that demonstrate the utility of entropy for quantification of diverse facets of human subjects’ cognitive ability and its effectiveness for capturing the effect of psychedelic drugs while studying such disorders [18,60].

The entropic measures have also appeared in the literature pertinent to meditation [204] and mediated social communication [205]. Although these are potentially interesting findings, they are primarily based on the association between observed changes in brain signal variability and the individuals’ subjective self-reported mental states. In this respect, a study by MacDuffie et al. [206] that included 1256 human subjects concluded that such self-report ratings were unrelated to measured neural activation. The effect of this shortcoming can be indeed observed in meditation and mediated social communication studies. For instance, in the case of mediation, whereas Vyšata et al. [204] reported a decrease in global entropy during the mediation (interpreted as a sign of relaxation by the authors), Kakumanua et al. [207] observed its increase which was only present in the case of experienced meditators. On the other hand, the mediated social communication studies assumed both, lower [205] and higher [208,209] entropy to indicate such self-assessed feelings as relaxed mood, interest, and the feeling of human presence. Furthermore, the mediated social communication studies were based on a surprisingly limited number of brain sites (e.g., only two forehead sites in [205,208,209]). As a result, it was unclear whether the observed changes in entropy were due to the individuals’ interaction with the media or potentially a residue of overall brain signal variability [1,3] which may not necessarily pertain to the effect of stimuli. Most importantly, their results were inconsistent with the findings that pinpointed the substantial contrast between human–human and human–agent interactions (HHI and HAI, respectively) [210,211,212,213]. Therefore, future research to clarify these discrepancies is necessary to allow for robust interpretation of the potential change in brain signal variability in response to such stimuli as meditation [204] and mediated social communication [205].

The entropic measures primarily adapted by the studies covered in the present overview included differential entropy (DE) [48], permutation entropy (PE) [214], and sample entropy (SE) [215] that forms the computational backbone of multiscale entropy (MSE) [72] (see Appendix B). There is an interesting underlying computational similarity between these measures on the one hand and the classical Shannon entropy [22] on the other hand. Specifically, whereas the original Shannon entropy was proposed for the case of discrete random variables, the computation of DE, PE, SE, and MSE are based on estimates of the entropy of a continuous time series through its discretization. Furthermore (although in a much-restricted sense), DE and PE both relate to SE computation through realization that the latter is based on a discretized (a binary discretization in this case) summary statistics of a given continuous time series. In this respect, it would be interesting for the future research to further examine the effect of such discretization strategies on the level of correspondence between the estimated entropy by these algorithms. Another possibility for future research that is worth investigating would be to reevaluate the use of PE for estimation of the brain variability based on its more recent variant, i.e., multiscale permutation entropy (MPE) [45].

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
