# Peer review of "Entropy and the Brain: An Overview"

_entropy, 2020, doi:10.3390/e22090917_

Round 1
Reviewer 1 Report
Summary:
The author proposes to review developments relating measures of entropy to brain function. The overview focuses on the role of entropy in three neural phenomena: consciousness, aging, and information processing.
Overall, the article provides a useful contribution to the literature linking information theory and neuroscience. However, there are two main ways that the article can be improved. In what follows I describe these two major points before listing a number of minor suggestions.
Major point 1:
The primary purpose of the article is to review applications of entropy to the study of brain function. However, the author spends little time actually defining entropy, either mathematically or intuitively. Therefore, in the main sections of the article, it is difficult to understand what exactly is meant by entropy in different contexts and how entropy has actually advanced our knowledge of the field.
For example, the author frequently refers to “variability” and “complexity” in a way that suggests that these concepts are related to entropy. However, because entropy is never properly defined or described, the links between these concepts remain unclear to the reader.
The main sections and conclusions of the article could be improved by a more in-depth introduction and discussion of entropy. This discussion (perhaps in its own section just after the introduction) could introduce entropy and give intuitive explanations for how entropy is related to the ideas of variability and complexity that are discussed throughout the remainder of the paper.
Major point 2:
The author selects three interesting topics to cover: consciousness, aging, and information processing. However, it is not made clear to the reader why these three topics are chosen and how they relate to one another. Is there a particular reason why the author chose these three topics? For example, are they particularly promising candidates for the use of entropy as an analytic technique? Is there a fundamental concept linking these three topics that entropy can help shed light on? The author should clarify these points in the abstract and introduction in order to help the reader understand why these particular topics were chosen.
Minor points:
Building off of the second major point (above), the transitions between sections feels disjoint. Currently, the first sentences of each of the major sections (2, 3, and 4) jump directly into describing various technical results, and the final sentences are also devoted to describing specific results. The flow of the paper could be improved by devoting 1-2 sentences at the beginning and end of each section to give an introduction and summary of the results being presented and how they relate to the overall goal of the paper (linking entropy and brain function).
The authors provide a nice description of different entropy measures in Appendix A, yet they do not refer to this appendix at any point in the main sections (2, 3, and 4). It would be helpful if the author referred to the appendix when describing relevant scientific results.
Overall, the article does a good job of providing a thorough discussion of the scientific results that are relevant to each of the three topics (consciousness, aging, and information processing). However, there are a small number of times where a significant amount of text is devoted to describing a specific results in detail when the same results could probably be covered more succinctly. Two examples: First, the first paragraph of section 2.3 is devoted to describing the limitations of techniques not related to entropy. Given that the goal is instead to focus on the useful insights that entropy can provide, I think this paragraph can be shortened in favor of more emphasis on the positive contributions of entropy-based measures. Second, the final paragraph of the manuscript is quite long and focuses on a specific direction for future work. The conclusion of the article could be made much more decisive by shortening this paragraph and including 2-3 sentences that wrap up the entire paper.
Author Response
First and foremost, the author would like to take this opportunity to express his gratitude for the reviewer’s time and kind consideration in reviewing the present manuscript. Comments by the reviewer helped substantially improve the quality of the present overview and its presentation.
In what follows, point-by-point responses to the reviewer’s comments and concerns are provided.
Sincerely,
Reviewer 1
Major point 1:
Reviewer’s Comment: The primary purpose of the article is to review applications of entropy to the study of brain function. However, the author spends little time actually defining entropy, either mathematically or intuitively. Therefore, in the main sections of the article, it is difficult to understand what exactly is meant by entropy in different contexts and how entropy has actually advanced our knowledge of the field.
For example, the author frequently refers to “variability” and “complexity” in a way that suggests that these concepts are related to entropy. However, because entropy is never properly defined or described, the links between these concepts remain unclear to the reader.
The main sections and conclusions of the article could be improved by a more in-depth introduction and discussion of entropy. This discussion (perhaps in its own section just after the introduction) could introduce entropy and give intuitive explanations for how entropy is related to the ideas of variability and complexity that are discussed throughout the remainder of the paper.
Author’s Response: The author would like to thank the reviewer for pointing out this important issue that was not properly included in the first version of manuscript. To address the reviewer’s comment, the Appendix is extended to include a new section (Appendix A Entropy, lines 441-480, in current version of the manuscript).
Under the main heading (lines 441-447 and equation (A1), in current version of the manuscript), the “entropy” is formally defined. This is followed by three subsections:
- Appendix A.1 Principle of Maximum Entropy (lines 448-454, in current version of the manuscript). This subsection points out a major development in understanding and interpretation of entropy as it was introduced by Shannon. This progress that was mainly attributed to the findings by Edwin T. Jaynes argues that “among all possible distributions, the one that maximizes the Shannon information entropy must be preferred.” The section also highlights the utility of this principle for mathematical modeling of the brain’s neural activity. The content of this subsection reads as follows.
“A substantial insight about understanding and interpretation of Shannon entropy came through Edwin T. Jaynes who argued that among all possible distributions, the one that maximizes the Shannon information entropy must be preferred [237,238] (i.e., the principle of maximum entropy, see also Cover and Thomas [18, p. 410, Theorem 12.11] for a formal statement). The principle of maximum entropy has a demonstrated utility in further advancing the mathematical modeling of brain’s neural activity [45,75,239,240].”
- Appendix A.2 Information, Complexity, and Randomness (lines 455-460, in current version of manuscript). This subsection briefly describes the equivalence between such concepts as information, complexity, and randomness. Specifically, it highlights the advances that have been achieved through rigorous theoretical works by Solomonoff, Kolmogorov, and Chaitin. It also refers the interested readers to a fascinating and highly engaging non-technical work by James Gleick. This subsection reads as follows.
“A crucial (and perhaps quite often confusing/misleading) issue is the relation between such concepts as "information," "complexity," and "randomness." As the extensive works by Solomonoff [241], Kolmogorov[242], and Chaitin [243,244] has demonstrated, these are in fact equivalent when it comes to information. Interested readers can find a highly engaging non-technical treatment of this subject in Gleick [245, Ch. 12].”
- Appendix A.3 Entropic Interpretation of Brain's Function: A Simplified Example (lines 461-480, in current version of the manuscript). This subsection explains how the brain’s function might be interpreted in terms of change in its entropy. This section adapts a gross and oversimplified picture of how the brain may actually function. Nonetheless, such an abstract and oversimplification helps capture the essence of entropic interpretation of brain's function. The content of this subsection comprises a footnote and a symbolic representation of a simplified neuronal population. As a result, its content cannot be restated here in a readable way. The author requests the reviewer to refer lines 461-480, in current version of the manuscript for reviewing its content. Thank you.
Major point 2:
Reviewer’s Comment: The author selects three interesting topics to cover: consciousness, aging, and information processing. However, it is not made clear to the reader why these three topics are chosen and how they relate to one another. Is there a particular reason why the author chose these three topics? For example, are they particularly promising candidates for the use of entropy as an analytic technique? Is there a fundamental concept linking these three topics that entropy can help shed light on? The author should clarify these points in the abstract and introduction in order to help the reader understand why these particular topics were chosen.
Author’s Response: The reviewer’s comment is addressed through two modifications 1) content of the Abstract 2) content of Section 1.1 The Present Overview.
- The following content is added to the end of the Abstract (lines 19-25, in current version of the manuscript).
“The present study helps realize that (despite their seemingly differing lines of inquiry) the study of consciousness, the ageing brain, and the brain networks’ information processing are highly interrelated. Specifically, it identifies that the complexity, as quantified by entropy, is a fundamental property of conscious experience which also plays a vital role in brain's capacity for adaptation and therefore whose loss by ageing constitutes a basis for diseases and disorders. Interestingly, these two perspectives neatly come together through the association of entropy and the brain capacity for information processing.”
- Section 1.1 The Present Overview is modified to further explain why the consciousness, the ageing brain, and the brain networks’ information processing were selected as topics of the present overview. In this Section, two reasons are provided. 1) the use of entropy in these three fields of research presented highly promising results 2) more importantly, the use of entropy allowed for the realization that their lines of inquiry substantially overlap. The modified content of this Section reads as follows (Section 1.1 The Present Overview, lines 72-85, in current version of the manuscript).
“The present overview aims at complementing the aforementioned reviews by focusing on three domains of research: the (altered) state of consciousness, the ageing brain, and the quantification of the brain networks' information processing. The reasons why these three areas of research are covered by the present overview are twofold. First, the study of consciousness, the ageing brain, and the brain networks' information processing are among the most active fields of research, to the best of our knowledge, where the use of entropy has resulted in highly promising findings. Second (and perhaps more importantly), the use of entropy in these areas helped realize that their seemingly different lines of research are indeed highly intertwined and related to each other. On the one hand, a number of theses consider complexity as a fundamental property of conscious experience[62,63] where the evolution of brain's complexity manifests as an entropy-enhancing process [64]. On the other hand, there are growing evidence that identify the vital role of physiologic complexity in organisms' capacity for adaptation [71, p. 3 and 93-94] [27], thereby relating the age-related loss of complexity with diseases and disorders [60,61,65-67]. Interestingly, these two propositions substantially overlap through their association with the brain capacity for information processing [13,68-70].”
Minor points:
Reviewer’s Comment: Building off of the second major point (above), the transitions between sections feels disjoint. Currently, the first sentences of each of the major sections (2, 3, and 4) jump directly into describing various technical results, and the final sentences are also devoted to describing specific results. The flow of the paper could be improved by devoting 1-2 sentences at the beginning and end of each section to give an introduction and summary of the results being presented and how they relate to the overall goal of the paper (linking entropy and brain function).
Author’s Response: To address the reviewer’s comment, the following changes are applied.
First, an opening paragraph is added to each of the three main sections. They read as follows.
- Section 2. (Altered) State of Consciousness, lines 110-116, in current version of the manuscript.
“Classical approaches to the study of consciousness primarily focus on contribution of specific brain areas or groups of neurons [63]. However, such approaches fall short in realizing that the modulation of information does not necessarily involve change in brain's regional power and/or activity [74]. Such a shortcoming can be addressed by analyzing the brain's activity using entropic measures that are, by principle, designed for quantification of the amount information. This, in turn, allows for identification of the brain signal's complexity (Appendix A.2) as a fundamental property of conscious experience.”
- Section 3. The Ageing Brain, lines 214-219, in current version of the manuscript.
“Section 2 identified the substantial contribution of the brain's entropy in formation of a conscious experience. In the same vein, a growing number of recent studies also hint at the interplay between physiological complexity and the brain’s capacity for adaptation. This section highlights some of these findings that show the significant contribution of the brain's signal complexity to its development and also recognize the loss of such a complexity through the process of ageing as a major contributing factor to various age-related cognitive declines and deficiencies.”
- Section 4. Quantification of the Brain Networks' Information Processing, lines 280-284, in current version of the manuscript.
“Sections 2 and 3 highlighted the correspondence of brain's entropy to manifestation of various states of conscious experience on the one hand and the decline of the brain's function in terms of loss of such an inherent complexity through the process of ageing on the other hand. As the current section reveals, such a correspondence is not a coincident but a substantial relation that emerges from the association between the brain's inherent complexity and its capacity for information processing.”
Next, the first paragraph in Section 5. Concluding Remarks provides an overall summary of the reviewed literature. It reads as follows (lines 342-361, in current version of the manuscript).
“The present overview sought to bring together three areas of neuroscientific research in which the use of entropy for quantification of the brain activity has yielded promising results. These areas were the (altered) state of consciousness, the ageing brain, and the quantification of the brain networks' information processing. While doing so, it highlighted three main observations. First, it identified that the use of entropic measures for study of consciousness and its (altered) states led the field to substantially advance the previous findings. For instance, they helped verify the theoretical models that envisioned the brain's large-scale information sharing underlies its conscious processing [75], that DMN and DAT networks were crucially involved in such processes [83], and that the conscious brain's dynamics sustained patterns of long-range coordination which was substantially distinct from its anatomical connectivity [29]. These studies further validated the utility of entropic measures as potential biomarkers for the study of differential states of consciousness [119-122,134]. Second, it realized that the use of these measures for study of the ageing brain provided significant insights on various ways that the process of ageing may affect its dynamics and information processing capacity. For example, they identified that the brain signal variability was a critical feature of its function [140] whose loss by ageing may help explain its functional decline [152] and reduced information integration [156,164]. Third, it revealed that their utilization for analysis of the brain regional interactivity formed a bridge between the previous two research areas. This included the evidence for the emergence of functional boundaries from stronger interaction among subsets of brain's global connectivity [176], the involvement of DMN in its higher information processing [180], and its reduced functional connectivity by ageing [196,198,199].”
This is followed by summarizing how the reviewed articles in this overview shed light on inter-relatedness between the study of consciousness, the ageing brain, and the quantification of the brain networks' information processing. It reads as follows (lines 362-366, in current version of the manuscript).
“Collectively, these studies hinted at the brain's complexity as its fundamental property that underlies the manifestation of such phenomena as consciousness [62,63] and adaptability [71, p. 3 and 93-94] [27]. This, in turn, resulted in attribution of various brain-related deficiencies and disorders [60,61,65-67] to the decline of this inherent complexity. These findings appeared to converge on the utility of brain's complexity in its function and information processing [13,68-70].”
Reviewer’s Comment: The authors provide a nice description of different entropy measures in Appendix A, yet they do not refer to this appendix at any point in the main sections (2, 3, and 4). It would be helpful if the author referred to the appendix when describing relevant scientific results.
Author’s Response: Although this is an important point that is raised by the reviewer, it is quite impossible to do so, given the current structure of these sections. This is mainly due to the fact that the different types of entropic measures were quite as often used in each of these three areas of research which, in turn, would result in Appendix to be repeatedly referred to in each of these three sections. The manuscript provided a brief explanation for this issue (Section 1.1 The Present Overview, lines 94-104, in current version of the manuscript) which reads as follows.
“There were two ways to organize the sections of this overview: 1) to focus on types of entropic measures 2) to categorize their contents as per domains of application. The first choice would allow for highlighting the different formulations of entropic measures and their subsequent modifications. However, it could also make it more inconvenient for the readers to pinpoint the fields of research in which these measures yielded substantial results. On the other hand, the second option would allow for more tangibly bringing together the incremental findings in each area of research, while putting less emphasis on types of entropic measures that were utilized. To more efficiently appreciate the recent progress based on application of entropy in neuroscientific studies, the present overview adapted the second approach. For interested readers, it also provided a brief summary of three most commonly used entropic measures in Appendix B. These measures are (differential) entropy (DE) [18,49], multiscale entropy (MSE) [73], and permutation entropy (PE) [72].”
The author apologizes for the inconvenience that this resposne may have caused the reviewer and would try to find a possible way to make such references, if the reviewer find their inclusions necessary. Thank you.
Reviewer’s Comment: Overall, the article does a good job of providing a thorough discussion of the scientific results that are relevant to each of the three topics (consciousness, aging, and information processing). However, there are a small number of times where a significant amount of text is devoted to describing a specific results in detail when the same results could probably be covered more succinctly. Two examples: First, the first paragraph of section 2.3 is devoted to describing the limitations of techniques not related to entropy. Given that the goal is instead to focus on the useful insights that entropy can provide, I think this paragraph can be shortened in favor of more emphasis on the positive contributions of entropy-based measures. Second, the final paragraph of the manuscript is quite long and focuses on a specific direction for future work. The conclusion of the article could be made much more decisive by shortening this paragraph and including 2-3 sentences that wrap up the entire paper.
Author’s Response: The first paragraph of the section 2.3 (lines 182-193, in current version of the manuscript) is shortened by primarily focusing on major points that legitimized the need for introduction of a more robust hypnosis biomarker. Similarly, the length of (previously) final paragraph of the manuscript (lines 395-414, in current version of the manuscript) is reduced by primarily focusing why the current results based on entropy in such studies as mediation and mediated social communication are not reliable enough an require further investigation.
In addition, a new closing paragraph is added to the end of manuscript that highlights the potential similarities between DE, PE, and MSE that are most frequently utilized in the study of consciousness, the ageing brain, and the quantification of the brain networks’ information processing. In effect, this paragraph is meant to persuade the future research to further enhance the results obtained by these entropic measures while considering their potential underlying similarities. Its content reads as follows (lines 415-428, in current version of the manuscript).
“The entropic measures that primarily adapted by the studies covered in the present overview included differential entropy (DE) [49], permutation entropy (PE) [218], and sample entropy (SE) [229] that forms the computational backbone of multiscale entropy (MSE) [73] (see Appendix B). There is an interesting underlying computational principle between these measures on the one hand and the classical Shannon entropy [23] on the other hand. Specifically, whereas the original Shannon entropy was proposed for the case of discrete random variables, the computation of DE, PE, SE, and MSE are based on estimate of the entropy of a continuous time series through its discretization. Furthermore (although in much restricted sense), DE and PE both relate to SE computation through realization that the latter is based on a discretized (a binary discretization in this case) summary statistics of a given continuous time series. In this respect, it would be interesting for the future research to further examine the effect of such discretization strategies on the level of correspondence between the estimated entropy by these algorithms. Another possibility for future research that is worth investigating would be to reevaluate the use of PE for estimation of the brain variability based on its more recent variant i.e., multiscale permutation entropy (MPE) [43].”

Reviewer 2 Report
Major comments
This is a very nice review with many sources. It argues for itself well and does its job simply and professionally. I recommend publication.
However, this needs work throughout on the grammar and English. It is good but needs further editing to say things concisely, as well as make references clear, as well as it needs more brevity as it could be a couple hundred words shorter than it currently is. I suggest an edit for brevity and also cutting down on confusing or overly-long sentences. This is a minor change, though.
Examples of grammar/English
“information capacity of the working memory (WM) [24]”
No “the” before working memory
“of entropy for emotional states classification based”
No plural for states
Part 1.1 should not be called a “study” since it is a review article
English 70-76 the word “It” is used without clear reference in many cases. The same is the case throughout the manuscript.
Line 77 needs to be better English
“King et al. [64] demonstrated that the state of consciousness was associated with a markedly increased information sharing among distributed brain region”
No “a”
“at envisioned the brain large-scale”
Brain’s
“diagnose the consciousness-related disorders”
No “the”
Section 2.2 needs a better title with correct grammar
DMN needs to be introduced as an acronym
Author Response
First and foremost, the author would like to take this opportunity to express his gratitude for the reviewer’s time and kind consideration in reviewing the present manuscript. Comments by the reviewer helped substantially improve the quality of the present overview and its presentation.
In what follows, point-by-point responses to the reviewer’s comments and concerns are provided.
Sincerely,
Reviewer 2
Reviewer’s Comment: “information capacity of the working memory (WM) [24]”
No “the” before working memory
Author’s Response: “the” is removed (Section 1. Introduction, line 39, in current version of the manuscript).
Reviewer’s Comment: “of entropy for emotional states classification based”
No plural for states
Author’s Response: This is modified to read as follows (Section 1. Introduction, lines 44-45, in current version of the manuscript).
“of entropy for emotion classification based”
Reviewer’s Comment: Part 1.1 should not be called a “study” since it is a review article
Author’s Response: The title of Subsection 1.1 is changed to “1.1. The Present Overview” (line 71, in current version of the manuscript)
Reviewer’s Comment: English 70-76 the word “It” is used without clear reference in many cases. The same is the case throughout the manuscript.
Author’s Response: In this paragraph, the first reference to “it” is replaced with “the present overview.” The modified sentence reads as follows (Section 1.1. The Present Overview, lines 86-88, in current version of the manuscript).
“In the case of (altered) state of consciousness, the present overview highlights the utility of entropy for differentiating the brain activity in three states of wakefulness, anaesthesia, and unresponsive wakefulness syndrome.”
In addition, the entire manuscript has been audited and subsequently similar refences have been modified.
Reviewer’s Comment: Line 77 needs to be better English
Author’s Response: This sentence is modified as follows (Section 1.1. The Present Overview, lines 94-95, in current version of the manuscript).
“There were two ways to organize the sections of this overview: 1) to focus on types of entropic measures 2) to categorize their contents as per domains of application.”
Reviewer’s Comment: “King et al. [64] demonstrated that the state of consciousness was associated with a markedly increased information sharing among distributed brain region”
No “a”
Author’s Response: “a” is removed from this sentence (Section 2.1. Various Stages of (Un)consciousness, line 118-119, in current version of the manuscript).
Reviewer’s Comment: “at envisioned the brain large-scale”
Brain’s
Author’s Response: “brain large-scale” is changed to “brain’s large-scale” (Section 2.1. Various Stages of (Un)consciousness, line 128, in current version of the manuscript).
Reviewer’s Comment: “diagnose the consciousness-related disorders”
No “the”
Author’s Response: “diagnose the consciousness-related disorders” is changed to “diagnose consciousness-related disorders” (Section 2.1. Various Stages of (Un)consciousness, line 131, in current version of the manuscript).
Reviewer’s Comment: Section 2.2 needs a better title with correct grammar
Author’s Response: To more conveniently reflect the content of this subsection, its title is changed to “Anaesthesia”(line 172, in current version of the manuscript). Similarly, the title for Subsection 2.3. is changed from “Entropy and Hypnosis” to “Hypnosis” (line 181, in current version of the manuscript).
Reviewer’s Comment: DMN needs to be introduced as an acronym
Author’s Response: A list of abbreviations along with their full-forms (including DMN) is added to current version of manuscript (under the heading, Abbreviations, lines 437-440).

Round 2
Reviewer 1 Report
The author has done a good job of addressing my comments. The paper provides a good overview of topics at the intersection of information theory and brain function. It therefore gives a useful and important discussion of topics that are currently of interest to neuroscientists, complexity scientists, and information theorists. I therefore recommend publication.
Christopher Lynn
Reviewer 2 Report
It would have been helpful if the authors tracked changes on their manuscript. I can't tell what has been changed. Also it appears they did the minimum amount they could. I would have liked to see more voluntary changes and attempts to make it more readable. But still recommending for publication.